# A Parallel Algorithm for Matheuristics: A Comparison of Optimization Solvers

**Martín González \*, Jose J. López-Espín and Juan Aparicio**

Center of Operations Research, Univeristy Miguel Hernandez of Elche, 03202 Elche, Spain;
jlopez@umh.es (J.J.L.-E.); j.aparicio@umh.es (J.A.)
**\*** Correspondence: martin.gonzaleze@umh.es

**Abstract:** Metaheuristic and exact methods are one of the most common tools to solve Mixed-Integer Optimization Problems (MIPs). Most of these problems are NP-hard problems, being intractable to obtain optimal solutions in a reasonable time when the size of the problem is huge. In this paper, a hybrid parallel optimization algorithm for matheuristics is studied. In this algorithm, exact and metaheuristic methods work together to solve a Mixed Integer Linear Programming (MILP) problem which is divided into two different subproblems, one of which is linear (and easier to solve by exact methods) and the other discrete (and is solved using metaheuristic methods). Even so, solving this problem has a high computational cost. The algorithm proposed follows an efficient decomposition which is based on the nature of the decision variables (continuous versus discrete). Because of the high cost of the algorithm, as this kind of problem belongs to NP-hard problems, parallelism techniques have been incorporated at different levels to reduce the computing cost. The matheuristic has been optimized both at the level of the problem division and internally. This configuration offers the opportunity to improve the computational time and the fitness function. The paper also focuses on the performance of different optimization software packages working in parallel. In particular, a comparison of two well-known optimization software packages (CPLEX and GUROBI) is performed when they work executing several simultaneous instances, solving various problems at the same time. Thus, this paper proposes and studies a two-level parallel algorithm based on message-passing (MPI) and shared memory (Open MP) schemes where the two subproblems are considered and where the linear problem is solved by using and studying optimization software packages (CPLEX and GUROBI). Experiments have also been carried out to ascertain the performance of the application using different programming paradigms (shared memory and distributed memory).

**Keywords:** parallel algorithm; exact methods; Mixed Integer Problems; MILP decomposition; matheuristics

## 1. Introduction

Mixed Integer Linear Programming (MILP) models deal with mathematical optimization linear problems involving two families of variables: discrete and continuous. The computation related with this problem is high when the number of variables increases.

Mixed Integer Linear Programming (MILP) problems are well-known in the literature of mathematical programming and they have been very useful for modelling classical problems in operations research (knapsack, inventory, production, location, allocation, scheduling, etc.). These classical problems, at the same time, are currently being used for making decisions in a wide variety of contexts (military, banking, business, etc.). Their great utility for modelling very popular problems is one of the reasons why many researchers have been interested in proposing different algorithms for solving MILP problems. Indeed, some of these problems, such as the famous Traveling

Salesman Problem (TSP), are combinatorial optimization problems classified as NP-hard and are therefore, difficult to solve when the size of the problem is large (see [1]).

In practice, two types of algorithms are used to solve MILP problems: exact methods and metaheuristic algorithms. On the one hand, the exact methods (for example, branch and bound, simplex, etc.) allow to determine the optimal solution of the problem being studied. In fact, these algorithms ensure optimality. However, they are not generally applicable for large problems and are known to be time-consuming for big or more complex databases. On the other hand, metaheuristics do not guarantee the optimality of the found solution but can be implemented when the instance to be solved becomes too large or difficult for exact methods. Two categories of metaheuristics are usually considered: single-solution algorithms (local search, tabu search, etc.) and population-based algorithms (evolutionary algorithms, swarm optimization, etc.) [2]. Nevertheless, a recent approach, called matheuristics that combines the two philosophies (Pradenas et al. [3], Li et al. [4]), i.e., exact methods and metaheuristics, has been proposed in the literature in an attempt to provide a more efficient solution method (see, for example [5,6]).

Matheuristic methodology has been introduced to find approximate solutions in a reasonable time for MILP problems (see [7]). In this paper, a parallel algorithm for a new matheuristic algorithm is proposed and studied taking into account an MILP-based decomposition [8], where the main problem is decomposed into two hierarchical subproblems where different families of optimization algorithms are used. This decomposition is based on the nature of the decision variables: continuous and discrete.

Overall, the matheuristic methods have been designed by investigating differing cooperation between metaheuristics and exact methods, to find the best combination to solve an MILP problem. A general classification of existing approaches combining exact methods and metaheuristics for MILP optimization is presented in [9].

The matheuristic algorithm studied in this paper uses an integrative combination, in which the metaheuristic provides information for the exact method, which solves the problem by providing new information to the metaheuristic. The main idea is to reduce the problem into much smaller subproblems which can be solved exactly by state-of-the-art mathematical programming algorithms. The variables and the constraints are partitioned into two sets, decomposing the main problem into two hierarchical subproblems: The metaheuristic fixes the decision variables in one set and the exact method optimizes the problem over the other set.

Some popular techniques found in the literature to solve these problems using decomposition approaches that exploit the problem structure [10] are studied, such as constraint decomposition approximation (cutting plane methods) [11] or inner approximation (Dantzig-Wolfe method) [12,13] and variable decomposition methods ([14,15]). Variable decomposition methods are considered in this paper.

There are different methodologies to design metaheuristics. Some design parameters determine the characteristics of each metaheuristic. Those parameters are framed into different search components: initialization, improvement, selection, combination and stopping criteria. Several methodologies [16] have been generalized to matheuristics, in which exact optimization is combined with a set of metaheuristics. Nevertheless, this paper does not focus in obtaining the best metaheuristic algorithm for the problem, but assumes that the best metaheuristic for the problem is known and focuses on the parallelization of the mathehuristic algorithm.

The parallel algorithms studied in this paper are based on message-passing and shared-memory paradigms since they are the most extended scheme of parallel algorithms in the literature. In the experiments, message-passing (MPI) [17] and OpenMP [18] are the two APIs used to developed the algorithms as they are extended library routines over C and C++.

Nowadays, usually HPC systems integrate both shared and distributed memory architectures. For such hybrid architectures, one can perform hybrid parallel programming by combining different parallel programming models that are used at different architectural levels within a single code. This can offer the program a greater degree of parallelism as well as better performance.



The NP-hard problem proposed requires the evaluation of a large space of solutions, which requires too much time and computation. Therefore, one way to reduce the cost is to divide this space into different sections, and explore them in parallel. Furthermore, the problem proposed requires evaluating numerous MILP problems to obtain a final solution, which, being independent of each other, can be evaluated in parallel. Thus, we have achieved two ways of optimizing computation time using two levels of parallelism: the problems to be evaluated, and the search space for solutions by problem.

In this paper, the main contribution is related with the throughput of a parallel algorithm solving a combination of MILP problems, using several optimization packages in the literature (CPLEX [19] and GUROBI [20]). We evaluate two parallel paradigms (MPI and OpenMP) to obtain the best configuration of resources that combines both paradigms by resorting to the use of two optimization software packages (CPLEX and GUROBI), obtaining the lowest time possible to solve a combination of MILP problems.

The organization of the paper is as follows. In Section 2.1, the proposed decomposition of MILP problems is presented. Section 2.2 details the matheuristic strategy combining linear continuous programming and discrete metaheuristics. In Section 3, we will focus on the parallel algorithm. Finally, Section 4 gives some computational experiments.

## 2. Related Work

### 2.1. MILP-Based Decomposition

Let us consider the following linear problem (LP):

$$\max \left\{ cx : Ax \leq b, x \geq 0, x \in \mathbb{R}^n \right\} \tag{1}$$

where $A$ is a $m \times n$ matrix, c a $n$-dimensional row vector, $b$ a $m$-dimensional column vector, and $x$ a $n$-dimensional column vector of continuous variables. If we add the restriction that certain variables must take integer values, a Mixed Integer Linear Program (MILP) appears, which could be described as follows:

$$
\begin{aligned}
& \max cx + hy \\
& s.t. \\
& Ax + Gy \leq b \\
& x \geq 0, x \in \mathbb{R}^n \\
& y \geq 0, y \in \mathbb{Z}^p
\end{aligned}
\tag{2}
$$

where $A$ is a $m \times n$ matrix, $G$ is $m \times p$ matrix, $h$ is a $p$ row-vector, and $y$ is a $p$ column-vector of integer variables.

An MILP problem is defined as a problem where discrete variables ($y$), which are restricted to integer values, and continuous variables ($x$), which can assume any value on a given continuous interval, are combined with integrality constraints.

Solving large-scale and complex instances using MILP techniques is not efficient in terms of search time. Indeed, large MILP problems are often difficult to solve by exact methods, due to the complexity of solving an optimization problem which includes integer and continuous variables. It is possible to solve large MILP problems by dividing them into smaller subproblems, and then solve them individually.

Figure 1 shows how a general MILP problem is decomposed into two hierarchical subproblems with different complexities:

- The subproblem (P1), which includes the discrete variables, is a computationally complex problem when it is large and an exact method is used for solving it. In this case, metaheuristic approaches could be more efficient than exact methods.

- The subproblem (P2), which includes the continuous variables, is a linear continuous problem that is easy to solve with exact methods.

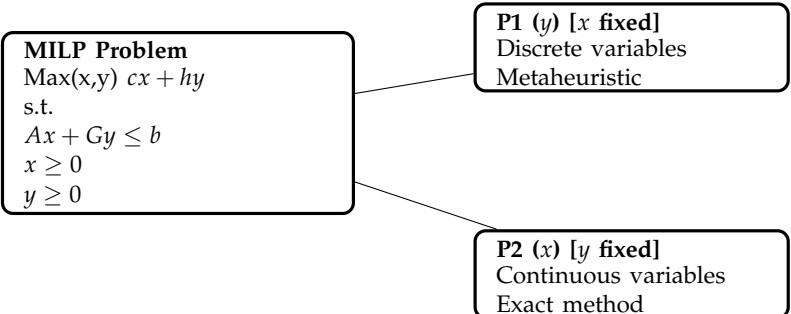

**Figure 1.** MILP problems decomposed into two subproblems.

## 2.2. Matheuristic Methodology

A matheuristic algorithm is defined as an algorithm which is made up of the combination of metaheuristics and exact methods. Thus, it is an optimization algorithm produced by the combination of metaheuristics and mathematical programming techniques. It can be the optimization of any mathematical problem which can be divided into different subproblems, where the main criteria are the nature of its variables.

In this work, a matheuristic algorithm is proposed to solve the problem shown in Figure 1, where the model is divided into two different subproblems, one of which is linear and the other discrete. The linear problem is easier to solve by exact methods, and the other subproblem can be studied using metaheuristic methods.

Whether or not the functions incorporated in the metaheuristic are executed (namely, initial population, improvement, selection, crossing and diversification), will depend on the selected metaheuristic. For example, an evolutionary algorithm (EA) does not use the improvement step, or a Greedy Randomized Adaptative Search Procedure (GRASP) does not use the crossover function. In this work, a parameterized scheme [21] that was previously studied by the authors in [16], is used to set the best metaheuristics.

Following the generation of the initial population by a metaheuristic, an exact method is involved to solve the subproblems generated. In this method, relaxation or decomposition techniques of the mathematical model are used. Relaxation methods consist of relaxing a strict requirement in the target optimization problem. This approach consists of ignoring the integrity constraints of an integer program and solving it using LP solvers. For that, the metaheuristic generates the decision variables and shares this information with the exact method.

Handling the constraints in the proposed decomposition methodology is a critical issue. The infeasible solutions generated by the exact method are evaluated and classified by assigning them a value based on certain parameters of the exact method. This parameter is related to the amount of restrictions that these solutions do not meet, and is modeled with a numeric value. This fitness penalty-based value is assigned to infeasible solutions. When this value is close to 0 it means that the solution is close to be a feasible solution. This implies that it requires fewer changes than other infeasible solutions to reach the feasible search.

A certain number of elements from both groups (feasible and infeasible solutions) are selected from the initial population and are used to generate new solutions through crossover and diversification functions. All these new generated solutions are also evaluated and improved in order to maximize the number of feasible solutions. When the algorithm uses the exact method, only the discrete variables in P1 can be used in the crossover function. The new variables generated are used to obtain a new solution solving P2 by the exact method. Those steps of the algorithm are repeated in a given number of iterations. Algorithm 1 shows the scheme of the main matheuristic algorithm. This algorithm

defines the parts involving the metaheuristic and the exact method. Every time that a model must be solved, the exact method is included, but if the model does not need to be solved, and only one solution is going to be evaluated, the metaheuristic is able to work alone.

---

**Algorithm 1:** Parallel matheuristic algorithm

---

**input** : $Problems(x, y)$
**output**: Best solution in each of the problems

*Fix the metaheuristic parameters;*
**for** $k = 1$ **to** $TotalProblems$ **do in parallel**
 *//Create $S_k$ set of solutions for problem k-th;*
 **for** $j = 1$ **to** $Population_k$ **do in parallel**
  *Fix discrete variables $vd_j$ of problem P1;*
  *Obtain continuous variables $vc_j$ solving P2 through exact method;*
  $S_k \leftarrow [Solution_j := (vd_j, vc_j)];$
  **if** *$Solution_j$ is not feasible* **then**
   *Improve $Solution_j$ using the best neighbourhood algorithm;*
  **end**
 **end**
 *not EndCondition //Select $SS_k$ subset of $S_k$ such as $|SS_k| > 1$;*
 **for** $w = 1$ **to** $Combination_k$ **do in parallel**
  *Select $s_1, s_2 \in S_k$ randomly ;*
  *Combine $vd_1$ and $vd_2$ the discrete variables of $s_1$ and $s_2$ saving as the discrete variables of a new*
   *solution $s_w$;*
  *Obtain $vc_w$ continuous variables of $s_w$ solving P2;*
  **if** *$Fitness(w) > Fitness(r_1)$ & $Fitness(w) > Fitness(r_2)$* **then**
   $SS_k \leftarrow s_w;$
  **end**
 **end**
 *//Improve $SS_k$ subset of $S_k$;*
 **for** $w = 1$ **to** $Improve_k$ **do in parallel**
  *Select $s_w \in SS_k$ randomly ;*
  **REPEAT**:;
  *Modify $vd_w$ using the best neighbourhood algorithm and obtain $vc_w$ solving P2 through exact*
   *method;*
  **UNTIL** *$Fitness(s_w)$ increase achieve EndConditions;*
 **end**
 *//Diversify $SS_k$ subset of $S_k$;*
 **for** $w = 1$ **to** $Diversification_k$ **do in parallel**
  *Select $s_w \in SS_k$ randomly ;*
  *Modify randomly $vd_w$ of $s_w$;*
  *Obtain $vc_w$ solving P2 through exact method;*
 **end**
 *Include $SS_k$ in $S_k$;*
 *$BestSolution_k \leftarrow s \in S_k$ such as $Fitness(s) \geq Fitness(w) \forall w \in S_k$ ;*
**end**

---

## 3. Parallel Algorithm

The parallel algorithm proposed is schemed in two levels. The first one considers the solutions of different independent MILP problems at the same time and the second one, the execution of each

solution of the mathehuristic algorithm. Thus, this scheme has the advantage of distributing the processors depending on the number of independent problems which must be solved and the number of solutions proposed for the matheuristic algorithm. Figure 2 shows the proposed scheme.

A set of solutions is created in each problem. Each solution contains discrete and continuous variables solving the two problems described in Figure 1. The discrete variables are set and the continuous variables are obtained using the discrete variables and lineal programming. When the initial set of solutions $S_k$ is created, a number of valid and invalid solutions, selected randomly from the reference set, are improved. A valid solution is considered when the continuous variables are obtained and the problem is feasible. Other solutions (infeasible ones) are considered invalid. The algorithm works with valid and invalid solutions improving the fitness value from the first one, and converts invalid solutions into valid ones.

The improvement function is developed following the variable neighbourhood search. It explores distant neighbourhood solutions of the currently selected solution. If an improvement is made, the old solution is replaced by this new one. The local search method is applied repeatedly to improve the selected solutions in the neighbourhood, obtaining local optima. At this point, a certain number of the best valid and invalid solutions are selected from the reference set $S_k$, creating a new subset $SS_k$. A combination function is applied to these selected solutions. It combines pairs of solutions randomly chosen from those previously selected. The combination is performed mixing the discrete variables using a binary mask, and then, the continuous variables are obtained through linear programming. Using this combination, new solutions are generated, but only the valid solutions are included in the reference set SSk.

In the implementation of the parallel algorithm, both shared memory (OpenMP) and message-passing (MPI) schemes have been proposed according to the following ideas: MPI is used in the first level of parallelization where the different independent MILP problems are considered. Therefore, the total number of MILP problems are distributed between the processor and the resources assigned to MPI. On the other hand, OpenMP is used in the internal algorithms. Then, the different parts of the matheuristic are parallelized using the processors and the resources assigned to OpenMP.

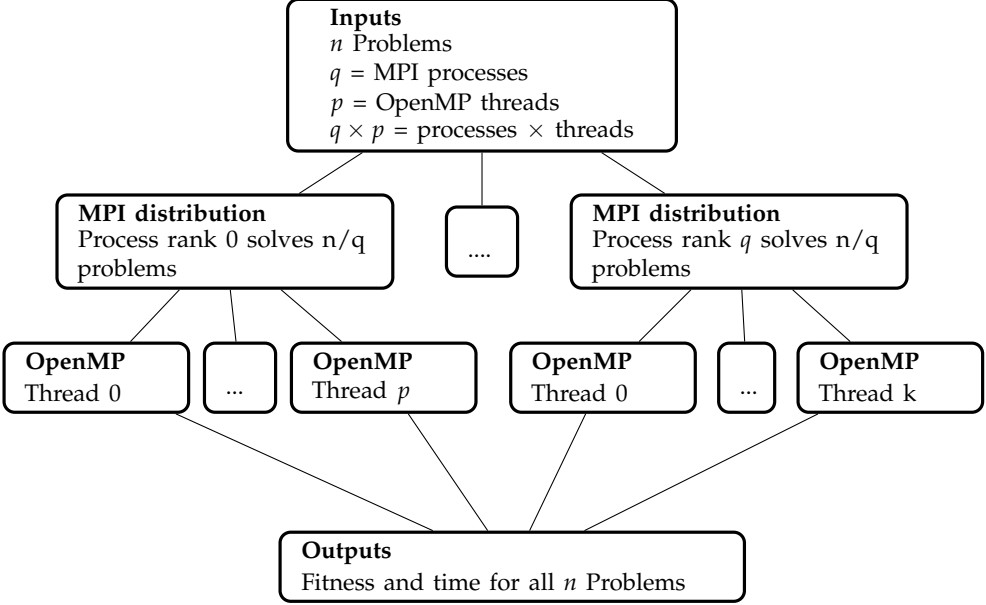

**Figure 2.** The parallel algorithm scheme based on resource decomposition in two levels.

In the literature, there are several mathematical methods that can solve both mixed integer linear programming and linear programming problems. In the algorithm developed in this paper, an MILP-based decomposition is used to divide the main problem, which is difficult to solve, into smaller LP-type problems that are easier to solve. In this regard, an exact method able of

optimally solving numerous LP problems is needed. For this task, two well-known optimization software packages, CPLEX and GUROBI, are evaluated to measure their performance in combination with our parallel algorithm.

## 4. Experimental Results

In this section, a computational experiment is carried out by applying the Algorithm 1 on an MILP problem associated with a modern Data Envelopment Analysis (DEA) technique. This is a non-parametric technique whose objective is to determine the technical efficiency of a set of $n$ firms (in general, a set of Decision Making Units—DMUs), which use $m$ inputs to produce $s$ outputs (see [22])

In particular, we focus our attention on the MILP problem proposed in [23] in the context of DEA. A formalization of the problem is described as follows. Let us assume that data on $m$ inputs and $s$ outputs for $n$ homogeneous DMUs are observed. For the $j$-th DMU, these are represented by $z_{ij} \geq 0$, $i = 1, \ldots, m$ and $q_{rj} \geq 0$, $r = 1, \ldots, s$. The DEA model that should be solved for evaluating the performance of DMU $k$ is as follows:

$$\max \left\{ \beta_k - \frac{1}{m} \sum_{i=1}^{m} \frac{t_{ik}^-}{z_{ik}} \right\}$$

s.t.

$$
\begin{aligned}
\beta_k + \frac{1}{s} \sum_{r=1}^{s} \frac{t_{rk}^+}{q_{rk}} &\leq 1 & & (c.1) \\
-\beta_k - \frac{1}{s} \sum_{r=1}^{s} \frac{t_{rk}^+}{q_{rk}} &\leq -1 & & (c.2) \\
-\beta_k z_{ik} + \sum_{j=1}^{n} \alpha_{jk} x_{ij} + t_{ik}^- &\leq 0 & \forall i & \quad (c.3) \\
\beta_k z_{ik} - \sum_{j=1}^{n} \alpha_{jk} x_{ij} - t_{ik}^- &\leq 0 & \forall i & \quad (c.4) \\
-\beta_k q_{rk} + \sum_{j=1}^{n} \alpha_{jk} y_{rj} - t_{rk}^+ &\leq 0 & \forall r & \quad (c.5) \\
\beta_k q_{rk} - \sum_{j=1}^{n} \alpha_{jk} y_{rj} + t_{rk}^+ &\leq 0 & \forall r & \quad (c.6) \\
-\sum_{i=1}^{m} \nu_{ik} z_{ij} + \sum_{r=1}^{s} \mu_{rk} q_{rj} + d_{jk} &\leq 0 & \forall j & \quad (c.7) \\
\sum_{i=1}^{m} \nu_{ik} z_{ij} - \sum_{r=1}^{s} \mu_{rk} q_{rj} - d_{jk} &\leq 0 & \forall j & \quad (c.8) \\
-\nu_{ik} &\leq -1 & \forall i & \quad (c.5) \\
-\mu_{rk} &\leq -1 & \forall r & \quad (c.6) \\
-d_{jk} &\leq -M b_{jk} & \forall j & \quad (c.7) \\
\alpha_{jk} &\leq M(1 - b_{jk}) & \forall j & \quad (c.8) \\
b_{jk} &= 0, 1 & \forall j & \quad (c.9) \\
-\beta_k &\leq 0 & & (c.10) \\
-t_{ik}^- &\leq 0 & \forall i & \quad (c.11) \\
-t_{rk}^+ &\leq 0 & \forall r & \quad (c.12) \\
-d_{jk} &\leq 0 & \forall j & \quad (c.13) \\
-\alpha_{jk} &\leq 0 & \forall j & \quad (c.14)
\end{aligned}
\tag{3}
$$

where $M$ is a positive big number. For this particular MILP problem, the vector of continuous variables $x$ consists of $(\beta_k, t_{ik}^-, t_{rk}^+, d_{jk}$ and $\alpha_{jk})$, while the vector of integer variables consists exclusively of $b_{jk}$.

In DEA, each DMU has an MILP problem to be solved. Regarding the data, in our simulations, the $m$ inputs and $s$ outputs of each of the $n$ DMUs are generated randomly but take into account that the production function that governs the production situation is the Cobb–Douglas function [24], which is well-known in microeconomics.

For all the experiments, the IBM ILOG CPLEX Optimization Studio (CPLEX) and the GUROBI Optimizer v.8.1 are used with a Free Academic License. The experiments are executed in a DELL PowerEdge R730 node with 2 Intel(R) Xeon(R) CPU E5-2650 v3 @ 2.30 GHz (Santa Clara, USA), with 20 cores (40 threads) at 2.4 GHz and 25 MB SmartCache memory. A comparative analysis of both optimization solvers in terms of capabilities showing different features of their architectures is shown in [25].

Experiments were performed to analyze the performance of the proposed algorithm using different parallelism tools. In addition, the performance of the different optimization packages when they are executed in parallel is analyzed. A fixed configuration of parameters for each metaheuristic function (population, combination, improvement, diversification and EndCondition in Algorithm 1) is used for conducting all the experiments. Parameters have been established based on previous analysis and experiences of the authors. Those parameters are: population = 310, combination = 49, improvement = 17, diversification = 10, EndCondition = 10 iterations (or five without improving).

In this paper, several DEA problems (model (3)) have been generated randomly with different sizes, so that the analyses are not dependent on the size of the problem. The problem sizes generated are the following:

- Size 1: $m = 3/n = 50/s = 1$
- Size 2: $m = 4/n = 50/s = 2$
- Size 3: $m = 5/n = 250/s = 3$

In the first experiment, the performance of the different optimization packages are analyzed. At this point, these optimization packages are evaluated solving several problems simultaneously; conflicts and problems of the optimization packages when they are executed simultaneously are also studied, using multiple instances at the same time through several threads.

Figure 3 shows the time that each optimization package takes for solving a single problem (in milliseconds) when, at the same time, there are different instances doing the same job when varying the number of processors. The time is always measured in processor number 0, for the execution of the same problem. At this point, just a single problem is executed (DMU 1 of n). It can be seen that, as more executions are performed simultaneously, the more the computing time increases. This is because the number of instances that can be simultaneously executed with these optimization packages are not unlimited. After a certain number of executions, the software starts to suffer delays. Analyzing the graph, it can be seen that from 10 simultaneous instances until 40, the computation time increases. This means that the improvement in computing time, when the number of resources increases, starts to decrease when using more than a number of processors which depends on the problem size.

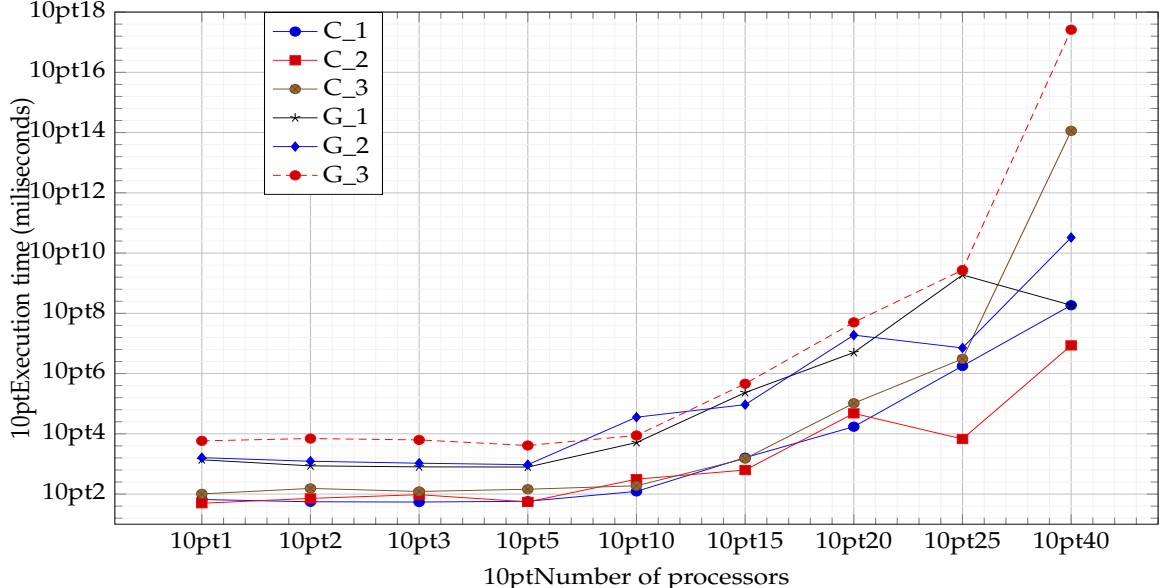

**Figure 3.** Computational time solving a unique LP problem in multiple processors using CPLEX and GUROBI solvers. The evaluated problems are: Size 1 with CPLEX (C_1), Size 2 with CPLEX (C_2), Size 2 with CPLEX (C_3), Size 1 with GUROBI (G_1), Size 2 with GUROBI (G_2) and Size 3 with GUROBI (G_3).

Another objective of the experiments is to compare how the proposed exact methods behave (CPLEX and GUROBI) according to the parallelization strategy. At this point, only the problem with size 1 ($m = 3$, $n = 50$ and $s = 1$) is analyzed in the following experiments. The results obtained are similar to the other sizes. Figure 4 shows the execution cost obtained when using the optimization packages and solving the problem shown in expression (3) with different paradigms of parallelism, solving all the DMUs included in the main problem. In this experiment, the parallelization levels (see Figure 2) have been executed separately. In the first instance, all the processors have been allocated to the highest level of parallelization (MPI), where the different problems to be solved among the available cores are divided. In the second instance, the processors have been allocated to the second level of parallelization (OpenMP). A previous work [26] shows that using OpenMP in the high level of parallelization is worse than using MPI and thus, this experiment is not considered in this paper. As a final result, a complete comparison of the levels of parallelism is presented, with the different proposed optimization packages in each level. It can be observed how, in terms of performance, a greater improvement in computing time is obtained using the second level of parallelism (OpenMP), and, in addition, when using CPLEX, the computational time is lower than when using GUROBI. However, as already mentioned, a convergence can be seen when more than 10 cores are used. This is because, despite the improvement due to the division of problems and the optimization of internal functionalities, by increasing the number of simultaneous instances, the optimization packages interfere with each other.

Once the effectiveness of the parallelism has been studied, the question is raised as to how to distribute the resources between the different parallel levels. For this, a third experiment has been developed to find the best configuration of available resources by dividing them between both parallel levels. The aim is find the optimal values for the variables $m$ and $k$ in Figure 2).

In this experiment, different possible configurations have been tested, obtaining, in each case, the time taken by the algorithm to find the optimal solution. Therefore, after some tests, we get the configuration that improves the best time shown in Figure 4. The time has been compared with that obtained using all the resources in both OpenMP and MPI, where all the available cores are focused on only one level of parallelism. These times are:

- OpenMP[CPLEX] = 43.1825 s
- OpenMP[GUROBI] = 74.3996 s
- MPI[CPLEX] = 119.4421 s
- MPI[GUROBI] = 159.6177 s

Table 1 shows the average of the time of 10 executions for each resources configuration. It is shown that computation time is minimized when resources are divided between different levels, compared to that obtained when all resources are provided at only one level. This proves that dividing the parallelism into different levels, and establishing the resources in an optimal way, the performance of the application improves. The experiments with lower times than those obtained by using MPI or OpenMP separately (shown above) have been highlighted in bold. In this way, it can be seen that in most of the cases, the cost using a mix parallel scheme is better than the cost of using just one of them. It can be observed that the configurations that allocate more resources to the shared memory level (OpenMP), such as combination 3-13 obtain better results than those that allocate more resources to the division of problems in distributed memory, such as configuration 13-3.

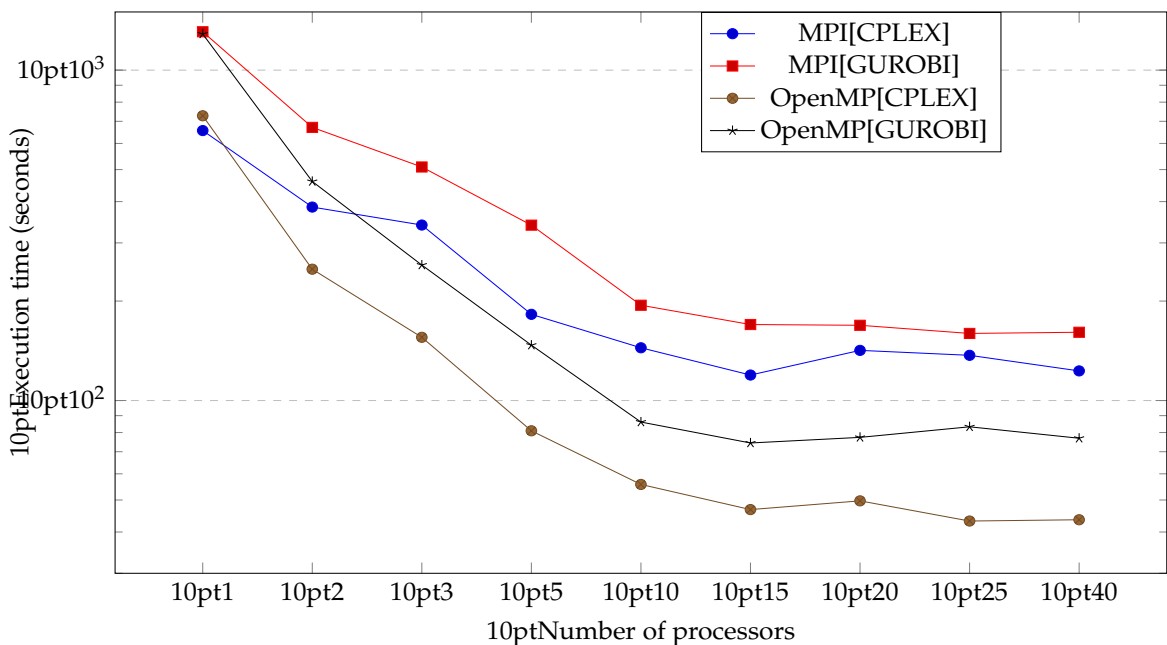

**Figure 4.** Performance of each level of parallelization. Comparison using all the resources with message-passing (MPI) or OpenMP with several optimization packages (CPLEX and GUROBI).

**Table 1.** Comparison between several configurations with MPI and OpenMP in a hybrid mode. Time for each solver is expressed in seconds. The number of processors used in each parallel level is also shown.

| Hybrid Parallel Configuration | | | | | | | |
|---|---|---|---|---|---|---|---|
| **GUROBI** | **CPLEX** | **MPI** | **OpenMP** | **GUROBI** | **CPLEX** | **MPI** | **OpenMP** |
| $223.7383_{16.4439}$ | $137.3334_{3.3673}$ | 2 | 2 | $104.2315_{8.9342}$ | $66.4368_{5.7361}$ | 3 | 3 |
| $81.3666_{5.8771}$ | $59.2839_{1.9190}$ | 4 | 3 | $59.6567_{2.1628}$ | $45.1889_{2.8821}$ | 4 | 5 |
| $\mathbf{68.5390_{3.6255}}$ | $51.4602_{3.0931}$ | 5 | 4 | $56.6961_{2.7679}$ | $\mathbf{40.6429_{1.5087}}$ | 3 | 8 |
| $\mathbf{53.2853_{2.2711}}$ | $\mathbf{42.5253_{1.9582}}$ | 5 | 8 | $\mathbf{70.2143_{6.3089}}$ | $54.2284_{2.3153}$ | 6 | 4 |
| $52.8286_{2.9948}$ | $40.0809_{2.4266}$ | 2 | 20 | $71.9496_{4.2597}$ | $75.3947_{4.6624}$ | 20 | 2 |
| $49.8065_{1.9282}$ | $37.3163_{2.4220}$ | 3 | 13 | $75.7056_{4.4348}$ | $62.6644_{2.2626}$ | 13 | 3 |
| $\mathbf{50.8675_{2.3173}}$ | $\mathbf{39.1378_{1.8670}}$ | 4 | 10 | $70.8639_{4.0900}$ | $56.7864_{2.6158}$ | 10 | 4 |

## 5. Conclusions and Future Works

In this paper, a parallel matheuristic algorithm for solving a set of Mixed Integer Linear Programming (MILP) problems is presented. The algorithm put forward follows a decomposition strategy proposed for large-scale MILP optimization problems. This decomposition is based on the nature of the decision variables (continuous versus discrete). In the proposed algorithm, an incomplete encoding representing only discrete decision variables is explored by metaheuristics. The encoding of solutions is completed for the continuous decision variables by solving a linear problem.

In the implementation of the parallel algorithm, both shared memory (OpenMP) and message-passing (MPI) schemes have been proposed in combination, in accordance with the following ideas: MPI is used in the first level of parallelization where the different independent MILP problems are considered and OpenMP is used for internal algorithms.

Experiment results are shown comparing cost and efficiency when processors and problem size vary, obtaining satisfactory results in terms of solution quality and execution time. The parallel study also focuses on the comparison of the CPLEX and GUROBI software packages since both are two of the most used packages when solving LP problems.

Finally, future research efforts could be aimed at determining the best metaheuristic, instead of assuming that the best parameters have already been found.

**Author Contributions:** M.G. and J.J.L.-E. designed the parallel model and the computational framework and analysed the data. J.A. designed the mathematical model. M.G. carried out the implementation and performed the calculations. M.G., J.J.L.-E. and J.A. wrote the manuscript with input from all authors. All authors have read and agreed to the published version of the manuscript.

**Acknowledgments:** The authors are grateful for the financial support from the Santander Chair of Efficiency and Productivity of the University Miguel Hernandez of Elche (Spain).

**Conflicts of Interest:** The authors declare no confict of interest.

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
