# Peer review of "A Parallel Algorithm for Matheuristics: A Comparison of Optimization Solvers"

_electronics, doi:10.3390/electronics9091541_

Round 1
Reviewer 1 Report
This paper presents a study on a hybrid parallel optimization algorithm for matheuristics and discusses the performance of different optimization software packages working in parallel, namely CPLEX and GUROBI optimization software packages. Overall, I found the material very interesting and well presented, and results well summarized in two graphs presented in figures 3 and 4.
However, I found the discussion related to the performance of the two optimization software packages presented on page 8, specifically from lines 238 to 252 a bit too vague. I believe it would be helpful to incorporate a little more information regarding the architecture of these two packages and try to reason around that.
Moreover, figures 3 and 4 should be improved. The font is too big (x and y labels and legends) compared to the main text, and the first letter in each label should be capitalized.
Author Response
This paper presents a study on a hybrid parallel optimization algorithm for matheuristics and discusses the performance of different optimization software packages working in parallel, namely CPLEX and GUROBI optimization software packages. Overall, I found the material very interesting and well presented, and results well summarized in two graphs presented in figures 3 and 4.
However, I found the discussion related to the performance of the two optimization software packages presented on page 8, specifically from lines 238 to 252 a bit too vague. I believe it would be helpful to incorporate a little more information regarding the architecture of these two packages and try to reason around that.
Done. A new reference and some new information have been added in the manuscript.
A comparative analysis of both optimization solvers in terms of capabilities showing different features of their architectures is shown in [25]
Anand, R.; Aggarwal, D.; Kumar, V. A comparative analysis of optimization solvers.Journal of Statisticsand Management Systems2017,20, 623–635.
Moreover, figures 3 and 4 should be improved. The font is too big (x and y labels and legends) compared to the main text, and the first letter in each label should be capitalized.
Done. The font of both figures has been modified.
Reviewer 2 Report
The authors introduce a hybrid parallel optimization algorithm for metaheuristics is studied. In this algorithm, exact and
5 metaheuristic methods work together to solve an MILP problem.
Figure 3, the x-axis looks strange for a scaling figure with the number of processors. Why did you choose these processor numbers? The same goes for the other figures with scaling.
Isn't it strange that in Figure 3, the solution time increases as you increase the number of processors?
Figure 4 and Table 1, what is the problem being solved?
Table 1 is confusing and needs a better explanation. Possibly being converted to a graph instead of a table.
I think there could be more figures and more explanations to show the problem in the current results and the advantages of the hybrid solution.
It is unclear to me why would you need to do MPI and OpenMP on top of it.
I would suggest the contributions of the paper be highlight more so it is easier to read.
Author Response
Comments and Suggestions for Authors
The authors introduce a hybrid parallel optimization algorithm for metaheuristics is studied. In this algorithm, exact and 5 metaheuristic methods work together to solve an MILP problem.
Figure 3, the x-axis looks strange for a scaling figure with the number of processors. Why did you choose these processor numbers? The same goes for the other figures with scaling. Isn't it strange that in Figure 3, the solution time increases as you increase the number of processors?
The number of processors has been selected with the objective of studying, firstly, the performance of parallelism in a few instances, and secondly, how it behaves by greatly increasing the number of processes.
Some new text (in red) has been added in the manuscript to clarify this point
Figure 3 shows simulated executions of individual problems of model 3. A specific DMU (in this case the number 1) has been selected and it has been executed simultaneously as many times as the processors used. As more executions are run simultaneously, the problem takes longer to solve. The time shown in figure 3 represents how much the optimizer takes to find the optimal solution. Theoretically, it should always take the same time, since it always solves the same problem, however the cost increase when the number of executions increases.
The computational time increases when the number of processors increases due to the fact that the system resources become congested when the number of instances of this type of software increases.
Figure 4 and Table 1, what is the problem being solved?
Done. Some text (in red) have been added (lines 247-249)
Another objective of the experiments is to compare how the proposed exact methods behave (CPLEX and GUROBI) according to the parallelization strategy. At this point, only the problem with size 1 (m=3, n=50 and s=1) is analyzed in the following experiments. The results obtained are similar to the other sizes. Figure 4 shows the execution cost obtained when using the optimization packages and solving the problem presented in expression 3 with different paradigms of parallelism, solving all the DMUs included in the main problem.
Table 1 is confusing and needs a better explanation. Possibly being converted to a graph instead of a table.
Done. We have added a new paragraph explaining in more details the Table 1.
The experiments with lower times than those obtained by using MPI or OpenMP separately (shown above) have been highlighted in bold. In this way, it can be seen that in most of the cases, the cost using a mix parallel scheme is better than the cost of using just one of them.
It can be observed that the configurations that allocate more resources to the shared memory level (OpenMP), such as combination 3-13 obtain better results than those that allocate more resources to the division of problems in distributed memory, such as configuration 13- 3.
I think there could be more figures and more explanations to show the problem in the current results and the advantages of the hybrid solution.
It is unclear to me why would you need to do MPI and OpenMP on top of it.
MPI is proposed to be used in the first level of parallelization because at that level of the algorithm the problems considered are independent, and thus, all the processors can work
together without sharing much information. Inside all of these problems and in the second level of the algorithm, Open-MP is used when sharing information is important, and thus, the sharing- memory parallel scheme works better. In the text, lines 178-183 focus on this idea.
I would suggest the contributions of the paper be highlight more so it is easier to read.
Done.
It has been added two new parts to the abstract to highlight the contributions of thepaper. Specifically, the red text shown next:
…In this algorithm, exact and metaheuristic methods work together to solve an MILP problem which is divided into two different subproblems, one of which is linear (and easier to solve by exact methods) and the other discrete (and is solved using metaheuristic methods). Even so, solving this problem has a high computational cost.
… is performed when they work executing several simultaneous instances, solving various problems at the same time. Thus, this paper proposes and studies a two-level parallel algorithm based on message-passing (MPI) and shared memory (Open MP) schemes where the two subproblems are considered and where the linear problem is solved by using and studying optimization software packages (CPLEX and GUROBI)

Round 2
Reviewer 2 Report
The authors wrote in their reply: "MPI is proposed to be used in the first level of parallelization because at that level of the algorithm the problems considered are independent, and thus, all the processors can work together without sharing much information. Inside all of these problems and in the second level of the algorithm, Open-MP is used when sharing information is important, and thus, the sharing- memory parallel scheme works better. In the text, lines 178-183 focus on this idea."
Again this raises the question of why not do all in Open-MP. Open-MP would not hinder or provide unnecessary overheads compared to MPI for the independent partition of a problem. This does not make sense to me.
Did you compare against a purely Open-MP problem?
Author Response
Using MPI in the high level of parallelization of the proposed algorithm works better than using OpenMP. We studied this comparison in a previous paper (Gonzalez et al., 2018) and this is the reason of why we don't consider this experiment in the current manuscript. Accordingly, we have added this reference and new information to the updated version of the manuscript in order to clarify this point. See the text on lines 251-255. González, M., López-Espín, J. J., Aparicio, J., & Giménez, D. (2018, June). A Parallel Application of Matheuristics in Data Envelopment Analysis. In International Symposium on Distributed Computing and Artificial Intelligence (pp. 172-179). Springer, Cham.